# Chemotaxonomic Identification of Key Taste and Nutritional Components in ‘Shushanggan Apricot’ Fruits by Widely Targeted Metabolomics

**DOI:** 10.3390/molecules27123870

**Published:** 2022-06-16

**Authors:** Bei Cui, Shu-Ming Liu, Tao Zheng

**Affiliations:** 1College of Forestry, Northwest Agriculture and Forestry University, Yangling, Xianyang 712100, China; cuibei@nwsuaf.edu.cn; 2College of Science, Northwest Agriculture and Forestry University, Yangling, Xianyang 712100, China

**Keywords:** ‘Shushanggan apricot’, sugar, organic acid, nutrients, metabolomics

## Abstract

The chemotypic and the content variation in taste substances and nutrients in ‘Shushanggan apricot’ fruits were detected by UPLC-MS/MS. A total of 592 compounds were identified, of which sucrose contributed mainly to the sweet taste and malic acid and citric acid were important organic acids affecting sweet–sour taste. γ-linolenic acid, α-linolenic acid and linoleic acid were the dominant free fatty acids, and neochlorogenic acid and chlorogenic acid were the predominant phenolic acids. Fruit taste was positively correlated with sucrose and negatively correlated with malic acid and citric acid. The differential metabolites were significantly enriched in the biosynthesis of amino acids and 2-oxocarboxylic acid metabolism pathways, regulating the sugar and organic acid biosynthesis. Taste and nutrient differences could be revealed by variations in composition and abundance of carbohydrates, organic acids and amino acids. The purpose of this study was to provide a comprehensive chemical characterization of taste and nutrient compounds in ‘Shushanggan apricot’ fruits.

## 1. Instruction

‘Shushanggan apricot’, belonging to the Prunus genus, Rosaceae family, has a unique flavor, intense aroma and abundant nutrition, and its flesh and almonds are favored by consumers due to the nutrients and phytochemicals. The fruits’ nutrition value mainly depends on its nutrition compounds [1,2]. In fact, fruit flavor not only reflects the differences in characteristic ingredients of fruit nutrients, but also renders consumer preferences [3]. ‘Shushanggan apricot’ fruits taste sweet and can be used to make jelly, fruit tea, enzymes, jam and fruit wine. ‘Shushanggan apricot’ fruits are also rich in phenolics, vitamins, amino acids, monounsaturated fatty acids and polyunsaturated fatty acids and other active ingredients, with significant antioxidation properties, which possess health benefits and can be used as basic materials for health food research and development.

The composition and content of the sugar, organic acid, carbohydrate, protein and other substances are the important indicators for evaluating fruit quality and have important influence on flavor and taste [3,4]. These traits also provide important information or sensory cues about the nutritional makeup of plant products [5,6]. Fruit taste is an important index to objectively reflect fruit quality, maturity and commodity quality [7], which directly determines the sensory quality and market acceptance of fruit and its processed products [8]. Sugars, amino acids, nucleotides, organic acids and other compounds are also essential basic substances and energy sources for plant growth, development and reproduction [9,10].

The sugar and organic acid constituents and their content in fruits are important economic and biological traits [11]. Fruit sweetness is determined by the content and composition of various sugars and sugar alcohols, and fruit acidity is mainly regulated by organic acids [12]. The organic acids display a declining trend during fruit development because they are the respiratory substrates consumed in glycolysis and TCA cycles [13,14]. Glucose and fructose are the main soluble sugar components in most fruits, while sucrose is the predominant sugar in peach, litchi and mandarin [15]. For organic acids, malic acid and citric acid occupy a dominant position in most fruits. Therefore, the investigation on the sugar and organic acid metabolism is conducive to illustrate the complicated metabolic process in fruit. The majority of fruit taste accumulation patterns have been investigated, such as tomato [16], strawberry [17], sweet orange [18], watermelon [19], pineapple [20] and grape [21]. The qualitative and quantitative analysis of fruit taste metabolites in biological samples was determined by metabolomics technology [22]. In addition, most of metabolomics studies concerning fruit tissues applied a combination of chromatographic separations coupled with mass spectrometric detectors [23]. The application of metabolomics technology in fruit quality research revealed the metabolic profiling variations in sugar and organic acids and other important nutrient substances (including anthocyanins, amino acids, free fatty acids and phenolic acids) closely related to the sugar and acid metabolism, which illuminated the synthesis and transformation regularity of these chemical components in fruit development [24,25]. However, the research on ‘Shushanggan apricot’ is mainly concentrated in seedling-raising techniques, cultivation and management techniques, stress resistance study and so on. The investigations on characteristics of key taste compounds, nutrients and active components in fruits have not been reported.

Here, the UPLC-MS/MS method was performed to investigate the primary metabolites in ‘Shushanggan apricot’ fruits at different developmental stages to reveal the composition and content differences in amino acids and their derivatives, organic acids, sugars and amino acids. The complete chemical characterization of taste substances and nutrients in ‘Shushanggan apricot’ fruits was established for the first time to understand the taste variations, which provided reference for the fruit taste quality evaluation, deepened the understanding of the characteristics of fruit nutrients and active ingredients and was conducive to the functional fruit product development.

## 2. Results

### 2.1. ‘Shushanggan Apricot’ Fruit Quality Index Analysis

Significant differences in flavor substances and nutrients (soluble solids, soluble sugar, organic acids, vitamin C, protein) were observed in ‘Shushanggan apricot’ fruits at the four development stages (immature green (G), color changing (CC I and CC II) and full mature (M)) (Table 1). The total soluble solid content (TSS) and SS (soluble sugar content) exhibited an increasing trend, while the titratable acidity values were on the downward trend. Vitamin C and protein contents were between 9.23 ± 0.02 and 25.56 ± 0.03 mg/100 g, 7.23 ± 0.19 and 18.63 ± 0.51 g/100 g, respectively. The sugar–acid ratio is the key factor affecting food taste and quality, which varied insignificantly at the early stage of fruit development and increased rapidly from color-changing to mature stages. The sugar–acid ratio at the mature stage was between 15 and 25, demonstrating that the taste was sweet–sour taste. The results revealed that the sugar–acid ratio was related to the rapid increase in SS and the rapid decrease in SS from the color-changing to mature stages.

The composition of sugar, organic acids and nutritional compounds in ‘Shushanggan apricot’ fruits at different development stages was significantly different. Clarifying the characteristics of nutrients and taste components in fruits was helpful for quality control and development of products with refined efficacy and strong functionality. Therefore, it is of great significance for qualitative and quantitative detection of taste and nutrients metabolites by widely targeted metabolomics analysis.

### 2.2. Metabolome Analysis in ‘Shushanggan Apricot’ Fruits at Developmental Stages

To investigate the differences between primary metabolites (flavor substances and nutrients) in ‘Shushanggan apricot’ fruits, UPLC-MS/MS analysis was carried out to detect the metabolite profiling at different development stages. A total of 592 primary metabolites were detected and identified in four developmental stages, including 68 organic acids, 66 free fatty acids, 16 vitamins, 65 sugars and alcohols, 43 nucleotides and their derivatives, 184 phenolic acids, 89 amino acids and their derivatives (Appendix A).

All biological replicates were grouped together (above the graph), indicating that the generated metabolic data were highly reliable (Figure 1A). Pearson correlation coefficients of each sample were greater than 0.99, indicating that the replicates within the group had strong correlation and good repeatability, which could be used for subsequent differential metabolites analysis (Figure 1B). In the PCA 3D map, the contribution rates of PC1, PC2 and PC3 were 46.75%, 23.68% and 12.40%, respectively, and the cumulative contribution rate reached 82.83% (Figure 1C). The trend of metabolite separation between groups was obvious, indicating that the fruits at different developmental stages had great differences at the metabolic level. Based on orthogonal signal correction and partial least squares-discriminant analysis (OPLS-DA) score plot (Figure 1D), the samples in each group were obviously separated. The OPLS-DA model compared the primary metabolite contents of the samples in pairs to evaluate the difference between G and CCI (R^2^X = 0.794, R^2^Y = 1, Q^2^ = 0.994), between CCI and CCII (R^2^X = 0.557, R^2^Y = 1, Q^2^ = 0.954), between CCII and M (R^2^X = 0.674, R^2^Y = 1, Q^2^ = 0.974) and between G and M (R^2^X = 0.8, R^2^Y = 1, Q^2^ = 0.995). The Q^2^ values of all groups were greater than 0.954 (Appendix A), indicating that the model was reliable and could be combined with VIP value analysis to screen differential primary metabolites.

### 2.3. Differential Metabolites Screening, Identification and Pathway Analysis

Based on OPLS-DA results, differential metabolites in different comparison groups were screened with variable importance in project (VIP) values ≥ 1, fold change ≥ 2 or fold change ≤ 0.5. The screening results were summarized in Appendix A. The number and variations in metabolites in comparison groups were clearly observed through volcanic plots. There were 272 significantly different primary metabolites between G and CCI (170 down-regulated, 102 up-regulated), 127 between CC I and CC II (59 down-regulated, 68 up-regulated) and 168 between CC II and M (121 down-regulated, 47 up-regulated) (Figure 2A–C). Furthermore, a Venn diagram was performed among these three combinations. As shown in Figure 2D, 36 common differentially accumulated metabolites (DAMs) were observed, and each comparison group had its unique differential metabolites (Appendix A).

Based on the KEGG database, the metabolic pathway enrichment of differential metabolites was investigated to understand the variation mechanisms of differential metabolites in the fruit development stage. The KEGG enrichment analysis demonstrated that the differential metabolites of G vs. CC I were significantly involved in glyoxylate and dicarboxylate metabolism, 2-oxocarboxylic acid metabolism, phenylpropanoid biosynthesis, linoleic acid metabolism and so on. For CC I vs. CC II, differential metabolites were allocated to the biosynthesis of secondary metabolites, biosynthesis of amino acids and purine metabolism. The differential metabolites of CC II vs. M were involved in the biosynthesis of cofactors, biosynthesis of amino acids, 2-Oxocarboxylic acid metabolism and phenylpropanoid biosynthesis, etc. Based on the statistical significance criterion corrected by multiple tests (adjusted *p*-value), phenylpropanoid biosynthesis, biosynthesis of amino acids, biosynthesis of secondary metabolites and 2-Oxocarboxylic acid metabolism pathways were significantly enriched (*p*-value ≤ 0.001) during ‘Shushanggan apricot’ fruit development (Appendix A).

### 2.4. Sugar, Organic Acids, Vitamin and Amino Acids

The dynamic variations in main sugar content in ‘Shushanggan apricot’ fruits at different development stages were shown in Figure 3A. The contents of sucrose kept sustainably increasing during fruit maturation, and the same trend was observed in dulcitol, D-sorbitol, L-fucitol, maltitol and allitol. The level of fructose, sucrose and glucose showed a rapid decrease from G to CC I, and a rise at the last three stages, and the variation patterns of inositol, 2-decanol and 1-decanol were similar with the above three main soluble sugars. Concerning sugar phosphates, most of them showed high accumulation at the CC I stage and rapidly decreased to a low level at the CC II stages. D-xylonic acid and gluconic acid were increased constantly throughout the developmental stages, while D-saccharic acid increased from G to CC I and decreased from CC II to M. These results showed that the sugars that determined the sweetness of ‘Shushanggan apricot’ fruits mainly accumulated from the CC II to M stage. The present study revealed that the dominant sugars in ‘Shushanggan apricot’ fruits were fructose, sucrose, glucose and D-mannose, and the sugar alcohols were dulcitol and D-sorbitol. Moreover, sucrose was the main sugar in ‘Shushanggan apricot’ mature fruits, demonstrating that the fruits belonged to sucrose-prevalent-type fruit.

Eight organic acids were detected in ‘Shushanggan apricot’ fruits, including α-ketoglutaric acid, succinic acid, isocitric acid, citric acid, quinic acid, malic acid, jasmonic acid and shikimic acid (Figure 3B). As expected, malic acid was the dominant organic acid, which revealed that the fruits belonged to malic-acid-dominant-type fruit. The content of citric acid, isocitric acid and malic acid increased rapidly from the early development stage G to the color-changing stage CC I and decreased rapidly from the color-changing stage CC II to the mature stage. The accumulation level of succinic acid also showed a significant downward trend with fruit development, and a rapid decrease from the green fruit stage G to the color-changing stage CC II. The level of α-ketoglutaric acid decreased with fruit development and ripening. Taken together, the organic acids mentioned above were the intermediates of the TCA cycle and they generally presented a distinct reduction in their levels during fruit development. However, the content of quinic acid and shikimic acid showed an increasing trend during fruit development, with a rapid decrease at CC II to M. The 2-methylsuccinic acid, methylmalonic acid, glutaric acid and aminomalonic acid evidently decreased at the early fruit development stage and accumulated to high levels at the mature stage. The content of 6-aminocaproic acid and 2-propylglutaric acid gradually decreased with fruit maturation, but in contrast, the γ-aminobutyric acid increased. For malonic acid derivatives, the contents of 2-isopropylmalic acid, 2-propylmalic acid and 3-isopropylmalic acid presented with a rapid decrease from the G to CC II stage and an increase from the CC II to mature stage. However, the content of 3-methylmalic acid continuously increased and accumulated to high levels at the mature stage. The present results demonstrated that malic acid citric acid and were the main organic acids in ‘Shushanggan apricot’ throughout the fruit development, which may contribute to the characteristic acidity taste in ‘Shushanggan apricot’.

A small number of the vitamins showed high accumulation levels at the maturity stage M, such as pyridoxal, pyridoxine, vitamin C, biotin and pyridoxine-5′-O-glucoside (Figure 3C). The majority of vitamins were presented with the similar variation pattern during fruit development: they accumulated high levels at the early fruit stage G, rapidly decreased at the color-changing stage and then slightly rose at the mature stage M, which included abundant common vitamins such as vitamin B3, isonicotinic acid, vitamin B1, N-(beta-D-Glucosyl) nicotinate, vitamin B2 and vitamin K2. The remaining vitamin compounds were highly accumulated at the G stage, were down-regulated at the last three stages and displayed the lowest levels at the mature stage, except for 4-pyridoxic acid.

The variations of 23 amino acids among fruit development were shown in Figure 3D. The content of most amino acids showed similar variation during fruit development: the accumulation was highest during the G stage then decreased rapidly in the color-changing and mature stages and the lowest content was presented at the mature stage. The amino acids in the above variation types were composed of L-Serine, L-Valine, L-Threonine, L-Leucine, L-Isoleucine, L-Norleucine, L-Asparagine, L-Ornithine, L-Homoserine, L-Methionine, L-Homomethionine, L-Histidine and L-Phenylalanine. A number of the amino acids showed a decreasing–increasing–decreasing trend during the development stages and presented the lowest contents at the color-changing stage CCI, such as L-Proline, L-Aspartic Acid, L-Glutamine, L-Lysine, L-Arginine and L-Tyrosine. The accumulation level of DL-Methionine and L-Allo-isoleucine increased to the peak point at CC I, and then decreased in the following stage to maturity. The L-Glutamic acid had the lowest level at the CC I stage and slightly increased at the CC II and M stages. However, L-Citrulline and L-Tryptophan were characterized by the increasing trend with high accumulation levels at the mature stage.

### 2.5. Free Fatty Acids and Phenolic Acids

Free fatty acids in ‘Shushanggan apricot’ fruits were mainly saturated fatty acids, containing a few monounsaturated fatty acids and polyunsaturated fatty acids (Appendix A). Among these free fatty acids, polyunsaturated fatty acids were the highest, which consisted of γ-linolenic acid, α-linolenic acid and linoleic acid. The monounsaturated fatty acids consisted of elaidic acid, palmitic acid, myristoleic acid and ricinoleic acid. For polyunsaturated fatty acids, the γ-linolenic acid and α-linolenic acid were highly accumulated in fruits at the mature green stage, and the contents of them were the lowest at the mature stage. The linoleic acid attained peak content at the mature stage and the lowest content at the color-changing stage CC I. For monounsaturated fatty acids, myristoleic acid, elaidic acid and palmitoleic acid were expressed with the highest level at the color-changing stage CC II. The variation in ricinoleic acid was an increasing–decreasing–increasing trend during fruit development, with the highest accumulation at the maturity stage. Among the saturated fatty acids, stearic acid, palmitaldehyde, petroselinic acid, 13-methylmyristic acid, undecylic acid, 12-hydroxyoctadecanoic acid, cis-10-pentadecenoic Acid (C15: 1) and palmitic acid presented small variations during fruit development. 13S-hydroxy-9Z,11E,15Z-octadecatrienoic acid, punicic acid, 9S-hydroxy-10E,12Z-octadecadienoic acid and 13(S)-HODE;13(S)-Hydroxyoctadeca-9Z,11E-dienoic acid presented the highest accumulation levels at G, decreased at the color-changing stages and then slightly increased at the mature stage. Myristic acid, 10-heptadecenoic acid, vaccenic acid, pentadecanoic acid and 13(s)-hydroperoxy-(9z,11e,15z)-octadecatrienoic acid showed a peak accumulation level at CCII and the lowest level at M. 13-Hydroxy-6,9,11-octadecatrienoic acid, 9,16-dihydroxypalmitic acid and lauric acid accumulated the highest levels at G, with the lowest level at M, while methyl linolenate and eicosadienoic acid accumulated high levels at the mature green stage, with quite a low level at CC I.

The changes in the most common phenolic acids during fruit development are illustrated in Appendix A. 3,4,5-trimethoxyphenyl-1-O-glucoside, salicylic acid-2-O-glucoside, ferulic acid, gallic acid were characterized by the increasing trend with high accumulation levels at the mature stage. 4-nitrophenol, vnilloylcaffeoyltartaric acid, 1-O-gentisoyl-D-glucoside, protocatechuic acid-4-O-glucoside, protocatechuic acid methyl ester, cimidahurinine, 1-O-salicyl-D-glucose and 5-(2-Hydroxyethyl)-2-O-glucosylphenol presented a similar variation pattern during fruit development, which were highly accumulated at the CC I stage and then rapidly decreased, whereas awsoniaside B, cichoriin, 1-O-eudesmoylquinic acid, 1-O-galloyl-D-glucose, cinnamic acid and androsin presented the highest accumulation levels at CC II. The remaining phenolic acids were presented with the highest content at the G stage, were down-regulated during the development stage and the variations were not obvious between the color-changing and mature stages.

### 2.6. Quantitative Detection of Key Taste Compounds by HPLC

The soluble sugars (fructose, glucose and sucrose) and organic acids (citric acid and malic acid) in ‘Shushanggan apricot’ fruits were quantitatively detected and verified by HPLC (Appendix A). Validation results showed that the accumulation of three soluble sugars and two organic acids at different developmental stages was significantly different, which was one of the main reasons for the different sugar–acid ratios, leading to different sweet or sweet–sour tastes (Figure 4A,B) and was consistent with the metabolomics results. In the early stage of fruit development, the content of total sugar, fructose, glucose and sucrose increased, but still maintained at a low level. In the late stage of fruit development, their content increased rapidly from the color-changing stage to the mature stage and reached the peak. Compared with glucose and fructose, the sucrose content was the lowest at the early fruit development stage and increased sharply with fruit maturation and became the major sugar in mature fruit. In contrast, malic acid and citric acid increased continuously from the immature green stage to the color-changing stage and decreased rapidly at maturity. In particular, the content of malic acid was higher than that of the citric acid during the whole development period and was observed as the main organic acid.

### 2.7. Correlation between Fruit Taste and Sugar and Organic Acids

Fruit taste is mainly determined by sugar, acid components and their content and proportion. To investigate the relationship between fruit taste and sugar and organic acid, correlation analysis was performed on the sugar and organic acid metabolites. As shown in Figure 4C, the total sugar content displayed significantly positive correlations with sucrose and sorbitol content and the strongest correlation with sucrose (r^2^ = 0.859), which was consistent with the proportion of different types of sugar content in total sugar, reflecting that sucrose was the predominant sugar in ‘Shushanggan apricot’ fruits. The total acid content presented high positive correlations with isocitric acid, citric acid and malic acid (r^2^ = 0.862, r^2^ = 0.898 and r^2^ = 0.932, respectively), indicating that the main components of organic acids in ‘Shushanggan apricot’ fruits were citric acid and malic acid, while malic acid accounted for about 60%. The results displayed that the acid–sugar ratio was significantly positively correlated with sucrose (r^2^ = 0.982) and negatively correlated with malic acid (r^2^ = −0.655) and citric acid (r^2^ = −0.617). Therefore, the sweet–sour taste was mainly controlled by sucrose, malic acid and citric acid, which also proved that those three compounds were the main sugar and organic acids in ‘Shushanggan apricot’ fruits. The above results revealed that the contribution rates of different sugar and organic acids to total sugar, total acid and sugar–acid ratio were different in ‘Shushanggan apricot’ fruits. The content of sugar and organic acids in fruit comprehensively affected the sugar–acid ratio and then affected the fruit taste.

## 3. Discussion

Previous studies on metabolites of ‘Shushanggan apricot’ mainly focused on specific categories of metabolites, such as sugar (fructose, glucose and sucrose). However, the overall differences in metabolic profiles of ‘Shushanggan apricot’ have not been thoroughly studied. The ‘Shushanggan apricot’ fruits at different developmental stages were selected as test materials to eliminate the complex influence of genetic background. In this study, UPLC-MS/MS-based widely targeted metabolite profiling analysis was used to preliminarily determine the composition and content characteristics of taste substances and nutrients, reveal the dynamic variations in sugar and acid during fruit development, deepen the cognition of the taste formation and provide theory for taste regulation of ‘Shushanggan apricot’ fruits.

In general, primary metabolites, especially sugars and organic acids, determined the fruit taste [26,27]. The accumulation of sugar, organic acids and other nutrient substances in fruit was regulated with the genetics and also influenced by cultivation conditions and environmental factors [28,29]. Nutrient concentrations and accumulation patterns in fruits were quite distinct, which was also regulated by fruit development [30]. Sucrose, glucose, fructose and sorbitol were the main soluble sugars, and malic acid, citric acid, succinic acid, tartaric acid and oxalic acid were the major organic acids in the *Cerasus humilis* fruits [31]. Sucrose, glucose and fructose were up-regulated slowly during the green and coloring stage and increased rapidly during the ripening period. Soluble sugars in ‘Shushanggan apricot’ fruits were the same as ‘Xinjiang apricot’, and glucose and sucrose were the main soluble sugars during fruit development [32]. Malic acid content was present with the trend of escalation, but the content was decreased before the fruits matured. The contents of citric acid, oxalic acid and succinic acid decreased with fruit maturation. During ‘Xinjiang apricot’ fruit development and ripening, the contents of malic acid and oxalic acid decreased dramatically, while the contents of citric acid, quinic acid and fumaric acid increased significantly, no clear consistent trend was observed for tartaric acid. The content of total acid in fruit increased during early development (from fruitlet to enlargement stage), but decreased rapidly during fruit ripening (from turning to full-ripe stage) [32]. Glucose and fructose significantly accumulate in sweet cherry during ripening. Fructose and glucose were the main sugars in the date plum persimmon fruit [33]. Our results revealed that malic acid and citric acid were the main organic acids in ‘Shushanggan apricot’ fruits, similar to peach fruits [34]. The ratios of fructose and glucose were higher in the early stage of fruit development, whereas sucrose accounted for a larger proportion in the late stage, indicating that the accumulation mode of sugar ‘Shushanggan apricot’ fruits was varied from glucose-predominated to sucrose-predominated during the development stage. The differences in organic acid composition and content led to fruits with unique acidic taste. The fruit taste is mainly determined by the acid–sugar ratio, and the acid–sugar ratio was positively correlated with total sugar content and sucrose content, indicating that the sugar acid ratio had the same trend with total sugar and sucrose content; moreover, the change rate of total sugar content was greater than that of total acid content. Based on the ratio of sugar and organic acid compounds, ‘Shushanggan apricot’ fruits belonged to sucrose-prevalent and malic-acid-dominant fruit.

Free fatty acids, having antioxidant, hypolipidemic, hypotensive and neuroprotective effects, can be used in food, cosmetics and pharmaceutical industries [35]. Unsaturated fatty acid is an important nutrient necessary for the human body and a fatty acid that constitutes fat in the human body [36,37]. There were abundant polyunsaturated fatty acids and monounsaturated fatty acids in ‘Shushanggan apricot’ fruits, including γ-linolenic acid, α-linolenic acid, linoleic acid, palmitic acid and ricinoleic acid. The relative content of polyunsaturated fatty acids was higher than that of monounsaturated fatty acids. Composition and abundance of amino acids are key indicators of nutritional quality and important factors determining taste. In total, 23 human essential amino acids were identified in ‘Shushanggan apricot’ fruits and accumulated differently during different developmental stages. The fruits were rich in essential amino acids, which can regulate immunity, brain health and blood pressure [38]. However, the relative content of most amino acids decreased with fruit development. The vitamin C content in mature fruits was the highest, which can promote the biosynthesis of bone collagen and is beneficial to wound healing and can enhance the immunity of the body [39]. Phenolic acids were one kind of phenolic compounds with good nutritional functions and pharmacological activities such as antioxidation [40,41]. Most phenolic acids had high contents at the early stage of fruit development and the color-changing stages. This study found that there were abundant bioactive nutrients in the ‘Shushanggan apricot’ fruits, which indicated that the fruits had the wide application foreground in health care functions.

In addition to the conventional detection of soluble sugars, organic acids and amino acids, KEGG enrichment analysis revealed three different metabolic pathways (phenylpropanoid biosynthesis, biosynthesis of amino acids, and 2-Oxocarboxylic acid) at different developmental stages, indicating that taste differences could be explained by changes in the composition and abundance of soluble sugars, organic acids, amino acids and phenolic acids. Alternatively, phenolic acid biosynthesis started from phenylpropionic acid biosynthesis pathway, these phenolic substances may be linked to flesh color, as flavonoids and flavonoids are pigments in many plants.

Through the analysis of the dynamic variations in primary metabolites in ‘Shushanggan apricot’ fruits at different developmental stages, the UPLC-MS/MS results indicated that ‘Shushanggan apricot’ fruits were rich in nutrients beneficial to the human body, especially vitamins, free fatty acids and phenolic acids. The detected substances could be developed into health food. Moreover, the increase in sucrose accumulation and the decrease in citric acid and malic acid during fruit development contributed to the fruits’ taste. Based on the variation trend in primary metabolites, the color-changing to mature stage was the key period of cultivation and management, and sufficient fertilizer and water should be provided to ensure nutrient and taste substance accumulation. Therefore, a comprehensive and deep understanding of the composition and variation characteristics of taste substances and nutrients in ‘Shushanggan apricot’ fruits would provide reference for fruit function development and further utilization.

In this study, the characteristic components of main metabolites in ‘Shushanggan apricot’ during fruit development were successfully identified by UPLC-MS/MS, and the variations in the composition and content of primary metabolites such as sugar, organic acids, amino acids and free fatty acids during fruit development were revealed. Further correlation analysis revealed the connectivity and relationship between sugar and organic acids and fruit taste, and fruit taste was positively correlated with sucrose and negatively correlated with malic acid and citric acid. Our results revealed that these compositional differences and concentrations of soluble sugars, organic acids, amino acids and phenolic acids may be the fundamental reason for controlling the taste of ‘Shushanggan apricot’ fruits. The detected substances can provide reference for the functional research and further development and utilization, important insights into taste-forming and nutrient compounds, and theoretical basis for the regulation of fruit taste quality of ‘Shushanggan apricot’.

## 4. Materials and Methods

### 4.1. Plant Materials and Chemicals

‘Shushanggan apricot’ fruits were harvested in the experimental plot located at Sanyuan, Shaanxi Province, China (34.5414° N, 108.8328° E). The young apricot fruits, born in the same inflorescence and with the consistent florescence were selected to be registered and marked (Figure 5). During the 2021 harvest season (from the 26 March to 14 June), the immature green (G), color-changing (CC I and CC II) and full mature (M) fruits were collected, and the fruit color could be distinguished at 29, 65, 72 and 81 days after flowering (DAF), respectively.

The fresh fruits, with uniform size, color and maturity, and without mechanical damage, were washed with distilled water, immediately put into liquid nitrogen and stored at −80 °C in the ultra-low temperature freezer for further use.

Chemical standards (fructose, glucose, sucrose, malic acid and citric acid) were purchased from Sangon Biotech (Shanghai) Co., Ltd.(Shanghai, China). Other reagents are analytically pure, purchased from Sinopharm Chemical Reagent Shanghai Co., Ltd.

### 4.2. Sample Preparation and Extraction

Four groups of test samples were subjected to vacuum freeze-drying (Scientz-100F) and ground (30 Hz, 1.5 min) to a powder by a mixer mill (MM 400, Retsch). We accurately weighed 100 mg powder, dissolved it with 1.2 mL 70% methanol solution, vortexed it 3 times and placed it in a refrigerator at 4 °C overnight. Following centrifugation at 12,000 rpm for 10 min, the supernatant was filtered by microporous membrane (SCAA-104, 0.22 μm pore size) and the filtrates were stored in the injection bottle before UPLC-MS/MS analysis.

### 4.3. Determination of Fruit Quality Indexes

After homogenizing the collected fruit flesh samples, a digital refractometer was used to determine the total soluble solid content (Brix %) based on the refraction method described by NY/T 2637-2014 (Atago PR-101R, Tokyo, Japan). Based on GB/T 12456-2008, the titratable acidity (%) was determined by NaOH acid-base titration method. Based on the Agricultural Standard NY/T 2742-2015, the total sugar (%) was measured by the anthrone-sulfuric method. Based on GB 5009.86-2016, the vitamin C content (mg/100 g) was determined by molybdenum blue spectrophotometry. Based on GB/T 5009.5-2003, the protein content (g/100 g) was detected using the kjeldahl nitrogen method. Each replicate contained 20 fruits and all determinations were performed in triplicate.

### 4.4. UPLC-MS/MS Analysis of Taste and Nutritional Compounds

Key taste and nutritional compounds in the ‘Shushanggan apricot’ fruit samples were extracted and analyzed using an UPLC-ESI-MS/MS system. The analytical conditions were as follows: (1) column, Agilent SB-C18 (1.8 µm, 2.1 mm × 100 mm); (2) mobile phase: solvent A was ultrapure water with 0.1% formic acid, and solvent B was acetonitrile acetonitrile with 0.1% formic acid; (3) gradient program: 0 min, 95:5 *v*/*v* (A/B); 0–9 min, 5:95 *v*/*v*; 9–10 min, 5:95 *v*/*v*; 11.1 min, 95:5 *v*/*v*; 11.1–14 min, 95:5 *v*/*v*; (4) flow rate, 0.35 mL per minute; (5) column temperature, 40 °C; and (6) injection volume, 4 μL. The effluent was alternatively connected to an ESI-triple quadrupole-linear ion trap (QTRAP)-MS. The ESI source operation parameters were as follows: (1) electrospray ion source temperature, 550 °C; (2) ion spray voltage (IS), 5500 V; (3) the ion source gas I (GSI), gas II (GSII) and curtain gas were set to 50, 60 and 25 psi, respectively; (4) the collision-activated dissociation (CAD) was high; (5) triple quadrupole (QQQ) scan using multiple reaction monitoring (MRM) experiments; (6) the collision gas (nitrogen) was set to medium; and (7) declustering potential (DP) and collision energy (CE) for individual MRM transitions were carried out with further DP and CE optimization. A specific set of MRM transitions were monitored for each period according to the metabolites eluted within the period.

Metabolite data analysis was conducted using the Analyst 1.6.3 software (AB SCIEX, Concord, ONT, Canada). Based on the OPLS-DA results and variable importance in projection (VIP) value, VIP ≥ 1 and fold change ≥ 2 or fold change ≤ 0.5 were set as the selection standard differential metabolites. Differential metabolites were mapped to metabolic pathways based on the Kyoto Encyclopedia of Genes and Genomes (KEGG) pathway database (*p*-values ≤ 0.05).

### 4.5. Quantitative Determination and Verification of Key Taste Compounds by HPLC

The contents of the main soluble sugars (sucrose, glucose and fructose) and organic acids (malic acid and citric acid) in different stages of ‘Shushanggan apricot’ fruits were determined by the HPLC method. The determination conditions of sucrose, glucose and fructose were as follows: (1) column, Angela Innoval NH_2_ (5 μm, 4.6 × 250 mm); (2) mobile phase, 85% acetonitrile aqueous solution; (3) flow rate, 0.9 mL/min; (4) column temperature, 40 °C; (5) injection volume, 20 μL; and (6) determination time, 40 min. Determination conditions of malic acid and citric acid were as follows: (1) column, GLWondaSil RC18 (5 μm, 250 mm × 4.6 mm); (2) mobile phase, 0.02 mol/L potassium dihydrogen phosphate solution and methanol (95:5); (3) pH of mobile phase, adjusted to 2.6 with phosphoric acid, ultrasonic degassing; (4) flow rate, 0.8 mL/min; (5) column temperature, 40 °C; (6) UV detector, 210 nm; and (7) injection volume, 8 μL. The contents of sugar and organic acids were calculated by peak area and standard curve (Appendix A) and expressed as mg·g ^−1^ FW.

### 4.6. Statistical Analysis

Principal component analysis, hierarchical cluster analysis and Pearson correlation coefficients were used for the multivariate statistical analysis of the metabolites. The analysis was carried out using R (http://www.r-project.org/, accessed on 26 May 2022). All samples were repeated three times and the results were expressed as mean ± standard deviation (SD). The significant difference was calculated by SPSS one-way ANOVA followed by Duncan’s test, and *p*-values < 0.05 were significant.

## Figures and Tables

**Figure 1 molecules-27-03870-f001:**
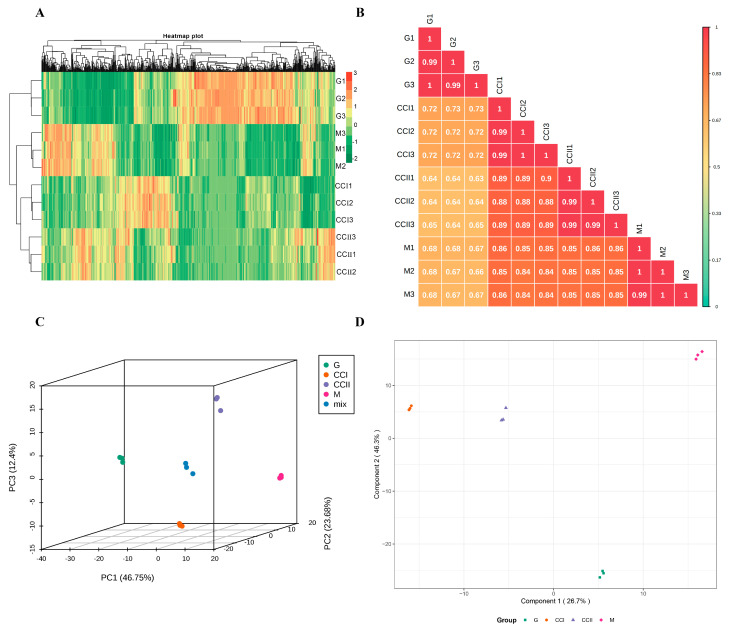
Differential fruit chemotype between ‘Shushanggan apricot’ fruits at four developmental stages. (**A**): Heat map of the identified metabolites; (**B**): Correlation analysis of four sets of samples; (**C**): Principal component analysis of metabolomics data (PCA 3D plot); (**D**): Orthogonal signal correction and partial least squares-discriminant analysis.

**Figure 2 molecules-27-03870-f002:**
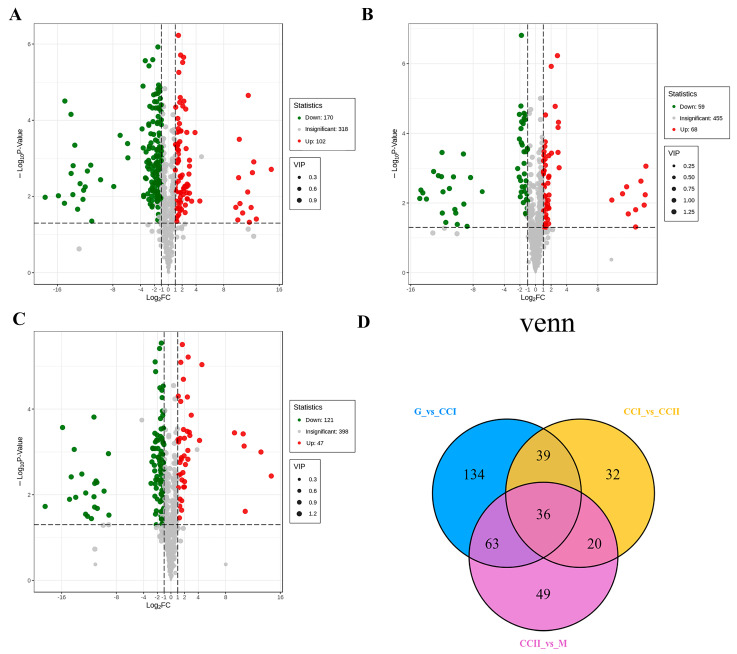
(**A**–**C**): Volcano plots of the different metabolites in the three comparison groups (G vs. CC I, CC I vs. CC II and CC II vs. M). (**D**): Venn diagram of differential metabolites among G vs. CC I, CC I vs. CC II and CC II vs. M.

**Figure 3 molecules-27-03870-f003:**
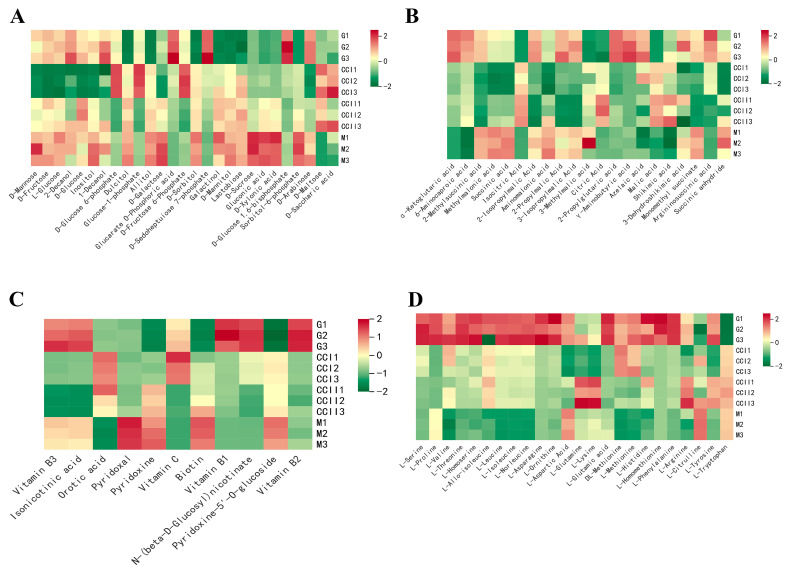
Primary metabolites detected by UPLC-MS/MS during four fruit developmental stages. (**A**): Saccharides and alcohols; (**B**): Organic acids (and its derivatives); (**C**): Vitamin; (**D**): Amino acids.

**Figure 4 molecules-27-03870-f004:**
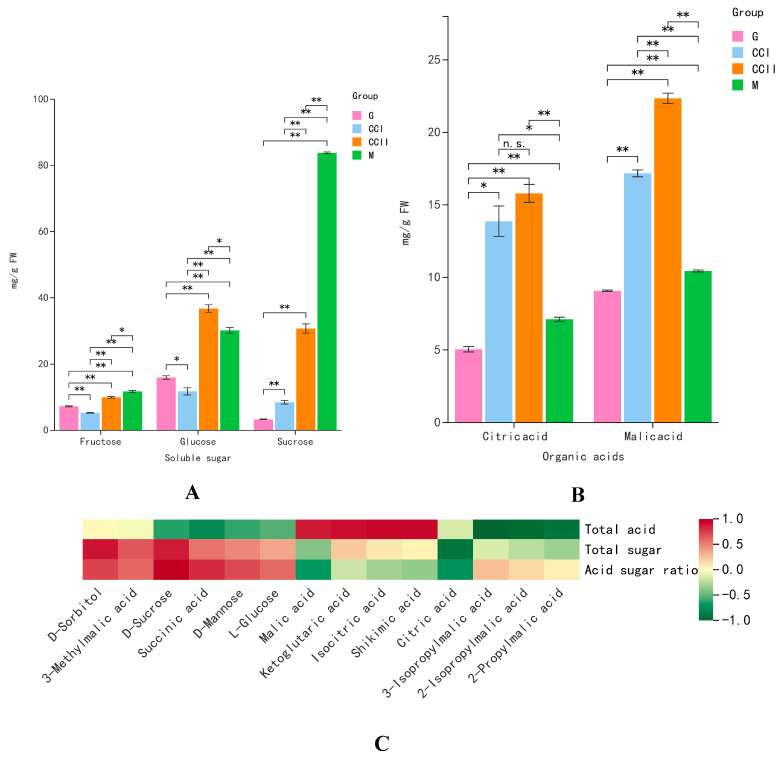
(**A**): The absolute content of soluble sugar (sucrose, fructose and glucose). (**B**): The absolute content of organic acid (citric acid and malic acid). (**C**): Correlation between fruit taste and sugar and organic acids during ‘Shushanggan apricot’ fruit development. The square color corresponds to the correlation value as shown in the legend: green represents a negative correlation and red represents a positive correlation. * represents significant difference at *p* < 0.05 level. ** represents significant difference at *p* < 0.01 level.

**Figure 5 molecules-27-03870-f005:**
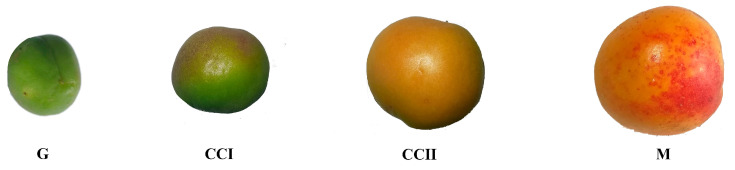
Morphology of ‘Shushanggan apricot’ fruits at different development stages: immature green (G); color-changing period (CCI and CCII); full mature (M).

**Table 1 molecules-27-03870-t001:** Soluble solid content, soluble sugar, organic acid, vitamin C and protein contents in ‘Shushanggan apricot’ fruit during development and ripening.

Development Stage	TSS (%)	TS (%)	TA (%)	VC (mg/100 g)	Pr (g/100 g)	ASR
G	15.56 ± 0.25 ^d^	6.25 ± 0.04 ^d^	0.89 ± 0.01 ^c^	11.42 ± 0.03 ^b^	7.23 ± 0.19 ^d^	7.00 ± 0.02 ^d^
CC I	20.21 ± 0.12 ^c^	8.58 ± 0.05 ^c^	1.06 ± 0.74 ^a^	9.23 ± 0.02 ^c^	11.27 ± 0.40 ^c^	7.18 ± 0.04 ^c^
CC II	25.95 ± 0.25 ^b^	10.61 ± 0.06 ^b^	1.20 ± 0.52 ^b^	20.45 ± 0.03 ^b^	15.13 ± 0.18 ^b^	10.10 ± 0.02 ^b^
M	29.14 ± 0.35 ^a^	12.59 ± 0.42 ^a^	0.80 ± 0.01 ^d^	25.56 ± 0.03 ^a^	18.63 ± 0.51 ^a^	16.84 ± 0.10 ^a^

Note: TSS (%)—total soluble solids; TS (%)—total sugars; TA (%)—titratable acidity; VC (mg/100 g)—vitamin C; PR (g/100 g)—protein; ASR—acid–sugar ratio. Each value was expressed as means ± standard (n = 3). Different lowercase letters between columns represent significant differences (*p* < 0.05).

## Data Availability

The data used to support the findings of this study are available from the corresponding author upon reasonable request.

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
