# Peer review of "Chemotaxonomic Identification of Key Taste and Nutritional Components in ‘Shushanggan Apricot’ Fruits by Widely Targeted Metabolomics"

_molecules, 2022, doi:10.3390/molecules27123870_

Round 1
Reviewer 1 Report
The main concern with the manuscript is that it lacks a detailed review of grammatical and syntactic errors, so before submitting it again, the authors need to perform an in-depth revision of the phrasing and formatting of the data. The authors should rewrite the results and discussion in separate paragraphs to ensure quality in the understanding of the manuscript. Throughout the document, there are impossible-to-interpret wordings, pleonasms, and redundant opinions when presenting the results. The units with which many of the results are presented are incorrectly described and are not specified in many parts of the manuscript. The abstract does not correctly reflect the results obtained in a scientifically correct manner. The discussion that is attempted to be presented between each results section is not scientifically sound because it lacks adequate wording. On the other hand, the raw data have potential, but because the redaction is so deficient, it complicates the correct reading and revision of the manuscript. I suggest a thorough revision of the writing and presentation of each data set before resubmitting the manuscript.
Author Response
Response: Those significant issues with the English (syntax, grammar, etc.) have been revised and marked in red in manuscript. We carefully modified the grammar and spelling of the article. We have adjusted the structure of the result part and rewrite the discussion part. In addition, the abstract was modified.
We tried our best to improve the manuscript and made some changes in the paper. These changes would not influence the content and framework of the paper. And here, we did not list the changes but marked in red in the revised paper.
We appreciated for your warm work earnestly, and hope the correction will meet with approval.
Reviewer 2 Report
In the manuscript entitled “Chemotaxonomic identification of key taste and nutritional components in ‘Shushanggan apricot’ fruits by widely targeted metabolomics” (molecules-1766578), Bei Cui, Tao Zheng, and Shu-ming Liu presented a study on the chemical characterization of taste and nutrients compounds in ‘Shushanggan apricot’ fruits. The authors performed UPLC-MS/MS analysis to determine primary metabolites in ‘Shushanggan apricot’ fruits at different developmental stages and established their chemical characterization of taste substances and nutrients. Overall, the manuscript was well prepared, and the experimental design was appropriate. However, there are several concerns in the manuscript that should be resolved before publishing it. Please consider the following comments to improve the manuscript.
1. The authors used MRM to analyze the metabolites (line 339). Therefore, the authors should include the MRM transition, DP, and CE of each metabolite. Importantly, the authors should clarify how to obtain these MRM transitions, DP, and CE. Were they done by using each metabolite standard?
2. The authors should not repeat the data in the main text when they are presented in tables. For example, in section 2.1, all the data from table 1 were repeated in the main text. It is unnecessary.
3. The quantification of key compounds was performed using HPLC-UV, while the LC-MS/MS was available with higher accuracy, precision, and sensitivity. Please clarify the reasons for that.
4. In the HPLC-UV method, please include the source and quality of standard metabolites.
5. Section 2.3 and Figure 2: the results should be considerably discussed. It seems that the authors ignore the discussion of this part. What can be inferred from the differential metabolites and KEGG pathways? From the hierarchical heatmap clustering analysis, the authors may determine some clusters that show changes in metabolite levels by developmental stages. The KEGG pathways identified from each cluster may provide insight into the development of the fruits.
6. The findings in this study should be compared or connected with those in previous studies, such as the profile of some metabolite groups and concentrations of key sugars
7. The authors should rewrite the method section. Many parts in this section were not complete sentences (e.g., lines 318-319, 323, etc.).
8. Figure 4: please clarify which statistical test was used.
9. Representative chromatograms of HPLC analysis of key sugars should be included in Supplementary materials.
10. Some abbreviations were not clarified before use (e.g., TSS – line 63, G, CCI, CCII, M – line 64, etc.).
11. Language editing is required to improve writing style and correct typos and grammar errors.
12. Line 177: typo (UPLC – MC).
Author Response
- The authors used MRM to analyze the metabolites (line 339). Therefore, the authors should include the MRM transition, DP, and CE of each metabolite. Importantly, the authors should clarify how to obtain these MRM transitions, DP, and CE. Were they done by using each metabolite standard?
Response: A specific set of MRM transitions were monitored for each period according to the metabolites eluted within the period.
- The authors should not repeat the data in the main text when they are presented in tables. For example, in section 2.1, all the data from table 1 were repeated in the main text. It is unnecessary.
Response: We have redescribed the fruit quality indicators to reduce duplication of table data. The total soluble solids content (TSS) and SS (soluble sugar content) exhibited an increasing trend, while the titratable acidity values were on the downward trend.
- The quantification of key compounds was performed using HPLC-UV, while the LC-MS/MS was available with higher accuracy, precision, and sensitivity. Please clarify the reasons for that.
Response: Based on UPLC-MS / MS detection, mass spectrometry data were processed by software Analyst 1.6.3, and the peak area (Area) of each chromatographic peak represented the relative content of the corresponding substances. The absolute content of key taste compounds was determined by HPLC method.
- In the HPLC-UV method, please include the source and quality of standard metabolites.
Response: The key flavor substances were detected by HPLC with glucose, fructose, sucrose, citric acid and malic acid as the standard substance.
- Section 2.3 and Figure 2: the results should be considerably discussed. It seems that the authors ignore the discussion of this part. What can be inferred from the differential metabolites and KEGG pathways? From the hierarchical heatmap clustering analysis, the authors may determine some clusters that show changes in metabolite levels by developmental stages. The KEGG pathways identified from each cluster may provide insight into the development of the fruits.
Response: Pathway enrichment analysis also identified significant differences in amino acids and 2-oxocarboxylic acid metabolism pathways during different development stages. Our results suggest that taste differences between the cultivars can be explained by variations in composition and abundance of carbohydrates, organic acids and amino acids. We have considerably discussed the KEGG enrichment in the discussion section.
- The findings in this study should be compared or connected with those in previous studies, such as the profile of some metabolite groups and concentrations of key sugars
Response: In the discussion section, we have added a comparison with previous research results.
- The authors should rewrite the method section. Many parts in this section were not complete sentences (e.g., lines 318-319, 323, etc.).
Response: We have revised and rewrite this method section.
- Figure 4: please clarify which statistical test was used.
Response: All samples were repeated three times, and the results were expressed as mean ± standard deviation (SD). The significant difference was calculated by SPSS one-way ANOVA followed by Duncan’s test, and p-values<0.05 were significant.
- Representative chromatograms of HPLC analysis of key sugars should be included in Supplementary materials.
Response: We have added the representative chromatograms of HPLC analysis of key sugars in Supplementary materials.
- Some abbreviations were not clarified before use (e.g., TSS – line 63, G, CCI, CCII, M – line 64, etc.).
Response: We have clarified the abbreviations of G, CCI, CCII, M (immature green (G), color-changing (CCâ… and CCâ…¡) and full mature (M)) and TSS (Total soluble solids content).
- Language editing is required to improve writing style and correct typos and grammar errors.
Response: Those significant issues with the English (syntax, grammar, etc.) have been revised and marked in red in manuscript.
- Line 177: typo (UPLC-MC).
Response: We have modified the ‘UPLC-MC’ to ‘UPLC-MS/MS’.
Round 2
Reviewer 1 Report
Line 29
Author Response
Thank you very much for your comments concerning our manuscript entitled “Chemotaxonomic identification of key taste and nutritional components in ‘Shushanggan apricot’ fruits by widely targeted metabolomics” (molecules-1766578). We appreciated for your warm work earnestly.
Reviewer 2 Report
The manuscript was revised according to previous comments. Some issues were suitably resolved. However, there are still several critical points as follows.
1. The triple quadrupole (QQQ) scan with multiple reaction monitoring (MRM) was used to identify each metabolite. The authors should include the MRM transition, DP, and CE of each metabolite (in supplementary materials).
2. The authors mentioned that "A specific set of MRM transitions were monitored for each period according to the metabolites eluted within the period”. It is not very clear. The authors should clarify how to obtain these MRM transitions, DP, and CE. Were they done by using each metabolite standard or obtained from a library? If there is no standard, how to ensure the reliability of the identification
3. Why was HPLC-UV used for the quantification of key compounds? Why not UPLC-MS/MS, considering that the standards and MRM methods were available? The LC-MS/MS has higher accuracy, precision, and sensitivity than HPLC-UV.
4. In the HPLC-UV method, please include the linearity range of each standard.
5. In Figure S4, please specify the concentration of each standard.
6. Line 353: please clarify “FW”.
Author Response
- The triple quadrupole (QQQ) scan with multiple reaction monitoring (MRM) was used to identify each metabolite. The authors should include the MRM transition, DP, and CE of each metabolite (in supplementary materials).
Response: The detailed information (Q1 (Da), Q3 (Da), Molecular Weight (Da), Formula Ionization model) of 592 metabolites was listed in Table S1.
- The authors mentioned that "A specific set of MRM transitions were monitored for each period according to the metabolites eluted within the period”. It is not very clear. The authors should clarify how to obtain these MRM transitions, DP, and CE. Were they done by using each metabolite standard or obtained from a library? If there is no standard, how to ensure the reliability of the identification
Response: Based on the self-built database MWDB (metware database), the material was qualitatively analyzed according to the secondary spectral information. Isotope signals, repetitive signals containing K+, Na+ and NH4+ ions, and repetitive signals of fragments of other larger molecular weight substances were removed. In MRM mode, the precursor ions (parent ions, Q1 (Da)) of the target substance were first screened by the quadrupole, and the ions corresponding to other molecular weight substances (Molecular Weight (Da)) were excluded to preliminarily eliminate the interference. Precursor ions (Q3 (Da)) break into a lot of fragment ions after induced ionization in the collision chamber. The fragment ions are then filtered by triple four-bar filter to select a characteristic fragment ion needed to eliminate non-target ion interference, so that the quantification is more accurate and reproducible.
- Why was HPLC-UV used for the quantification of key compounds? Why not UPLC-MS/MS, considering that the standards and MRM methods were available? The LC-MS/MS has higher accuracy, precision, and sensitivity than HPLC-UV.
Response: UPLC-MS/MS method was performed to detect the relative content of corresponding substances in MRM mode. The absolute contents of main soluble sugars (sucrose, glucose and fructose) and organic acids (malic acid and citric acid) in different stages of apricot were determined by HPLC method.
- In the HPLC-UV method, please include the linearity range of each standard.
Response: The detailed information of linearity range of 5 standard was listed in Table S5.
- In Figure S4, please specify the concentration of each standard.
Response: The detailed information of concentration of 5 standard was listed in Table S5.
- Line 353: please clarify “FW”.
Response: FW is the abbreviation of sample fresh weight.